# “Live-Autoradiography” Technique Reveals Genetic Variation in the Rate of Fe Uptake by Barley Cultivars

**DOI:** 10.3390/plants11060817

**Published:** 2022-03-18

**Authors:** Kyoko Higuchi, Keisuke Kurita, Takuro Sakai, Nobuo Suzui, Minori Sasaki, Maya Katori, Yuna Wakabayashi, Yuta Majima, Akihiro Saito, Takuji Ohyama, Naoki Kawachi

**Affiliations:** 1Laboratory of Biochemistry in Plant Productivity, Department of Agricultural Chemistry, Tokyo University of Agriculture, Tokyo 156-8502, Japan; ryukeikyou.34hi@gmail.com (M.S.); 3.1623memmey@gmail.com (M.K.); yuw.0104@icloud.com (Y.W.); ym.mrm.ap14@gmail.com (Y.M.); a3saito@nodai.ac.jp (A.S.); to206474@nodai.ac.jp (T.O.); 2Materials Sciences Research Center, Japan Atomic Energy Agency (JAEA), Ibaraki 319-1195, Japan; kurita.keisuke@jaea.go.jp (K.K.); sakai.takuro@jaea.go.jp (T.S.); 3Takasaki Advanced Radiation Research Institute, National Institutes for Quantum Science and Technology (QST), Takasaki 370-1292, Japan; suzui.nobuo@qst.go.jp (N.S.); kawachi.naoki@qst.go.jp (N.K.)

**Keywords:** barley, β-ray emitting nuclide, iron (Fe), Fe deficiency, phenotyping, photosystem, radioactive tracer, real-time imaging, transport

## Abstract

Iron (Fe) is an essential trace element in plants; however, the available Fe in soil solution does not always satisfy the demand of plants. Genetic diversity in the rate of Fe uptake by plants has not been broadly surveyed among plant species or genotypes, although plants have developed various Fe acquisition mechanisms. The “live-autoradiography” technique with radioactive ^59^Fe was adopted to directly evaluate the uptake rate of Fe by barley cultivars from a nutrient solution containing a very low concentration of Fe. The uptake rate of Fe measured by live autoradiography was consistent with the accumulation of Fe-containing proteins on the thylakoid membrane. The results revealed that the ability to acquire Fe from the low-Fe solution was not always the sole determinant of tolerance to Fe deficiency among barley genotypes. The live-autoradiography system visualizes the distribution of β-ray-emitting nuclides and has flexibility in the shape of the field of view. This technique will strongly support phenotyping with regard to the long-distance transport of nutrient elements in the plant body.

## 1. Introduction

Recently, non-invasive techniques for visualizing the location of radioisotopes in living plant bodies have been intensively developed for phenotyping. These techniques are expected to contribute to the evaluation of the capacity and characteristics of long-distance transport of nutrient elements in plants. Detection of ^11^C-labeled photo-assimilates, H_2_^15^O, or ^18^F-fluorodeoxyglucose using positron emission tomography (PET) is a powerful imaging technique for visualizing water and sugar flows [1]. The short half-life of these positron-emitting nuclides is advantageous for repetitive measurements in the same plant. Previously, carbon dynamics in plants were successfully visualized using the PlanTIS system [2], and “*pheno*PET” was further developed for high-throughput phenotyping [3]. Moreover, ^22^Na translocation in different plant species was visualized and characterized using a positron-emitting tracer-imaging system (PETIS) [4]. In further studies, clinical PET was used to monitor ^22^Na transport in plants [5], and ^52^Mn uptake and transport were compared in two maize genotypes using a mini-PET camera [6].

Chloroplasts have a high demand for iron (Fe) to perform photosynthesis [7], but the available Fe for root absorption is not always sufficient of plants under oxidative conditions [8]. Thus, the acclimation mechanism for improving Fe uptake has been studied extensively. Fe chelators that solubilize or stabilize Fe in aqueous solutions and many Fe transporters have been identified [9,10]. However, the actual Fe uptake rate has not been broadly surveyed in many species or genotypes using the same experimental system. We recently reported that the amount of acquired Fe may not always be the sole determinant of tolerance to Fe deficiency, and that acclimation of chloroplasts to low-Fe conditions also contributes to tolerance among barley cultivars [11]. Some Fe-deficiency-tolerant cultivars may exert tolerance depending on the effective Fe uptake, whereas others depend on Fe sparing. Thus, phenotyping of Fe uptake will provide knowledge for improving crop performance under Fe-deficient conditions. ^52^Fe translocation in barley cultivars was previously monitored using PETIS [12].

The positron-emitting nuclides mentioned above, except ^22^Na, are not commercially available, are produced using a cyclotron, and must be used immediately after generation because of their short half-life. Thus, it is difficult for this technique to be widely adopted by plant physiologists. In contrast, a real-time radioisotope imaging system (RRIS) was developed previously to facilitate dynamic imaging of commercially available β-ray emitting nuclides in living plants [13]. This system consists mainly of a scintillator and photon-counting camera. The β-rays emitted from the tracers were converted to light by a CsI scintillator, and the light was detected by a highly sensitive cooled charge-coupled device (CCD) camera. The dynamics of various β-ray-emitting nuclides in living *Arabidopsis* plants were visualized in a previous study using RRIS [14], and a new β-ray imaging system called “live-autoradiography” has been developed recently [15]. This system has the advantage of flexibility in the shape of the field of view, because it uses ZnS:Ag phosphor powder as the scintillator. In addition, because the scintillator of this system is installed inside a dark box, the distribution of β-ray-emitting nuclides in the plant body under continuous light conditions can be observed. Imaging experiments should be conducted under growth-light conditions because transpiration is active during the daytime and Fe loading to mesophyll cells and chloroplasts is facilitated under light conditions [16].

In this study, the rate of Fe uptake from a nanometer-order Fe solution was compared among the three barley cultivars using live-autoradiography. The system was equipped with a 100 × 100 mm imaging field, as originally described [15]. In the current study, two 200 × 200 mm fluorescent plates were used to improve the throughput. Moreover, the fluorescent material was replaced to obtain better luminescence because the uptake of trace elements, such as Fe, was low. The three barley cultivars used in this work were “Sarab 1 (SRB)” with very strong Fe-deficiency tolerance, “Ehimehadaka 1 (EHM)” with strong tolerance, and “Ethiopia 2 (ETH)” with weak tolerance [11]. SPAD values of EHM and ETH significantly decreased when cultivated with 3 μM Fe nutrient solution and ETH exhibited very severe chlorosis with 1 μM Fe, whereas SRB showed no obvious chlorosis even under the 1 μM Fe condition (Figure 1 in [11]). We show that one of the tolerant cultivars, EHM, showed excellent Fe absorption, whereas the cultivar SRB acquired a smaller amount of Fe than the Fe-deficiency sensitive cultivar ETH.

## 2. Results

### 2.1. Improvements in Live-Autoradiography for Phenotyping of Fe Uptake

Live-autoradiography has been previously reported to successfully detect ^37^Cs in a plant body [15], but the luminescence efficiency was not sufficient to detect smaller amounts of radioactive tracers in plants, which is necessary for future high-throughput applications. Therefore, the original system was modified prior to phenotyping. SiAlON (ceramics based on silicon, aluminum, oxygen, and nitrogen) phosphor powder, instead of ZnS:Ag, was used to improve the luminescence efficiency against β-rays for detecting small amounts of ^59^Fe in the plant body. Figure 1a shows images of a ^137^Cs point source with a diameter of 4 mm and radioactivity of 100 kBq acquired over 5 min using the ZnS:Ag and SiAlON scintillators. Dashed circles indicate the regions of interest (ROIs). The mean image intensities in the ROIs were 79.5 count (ZnS:Ag) and 123.0 count (SiAlON). The maximum energy of the main β-ray emitted from ^137^Cs was 514 keV, which was close to the ^59^Fe β-ray energy of 466 keV. Therefore, by changing the ZnS:Ag phosphor powder to SiAlON, the sensitivity can be improved by 1.5 times when measuring ^59^Fe.

The original system can obtain an image of only one plant at a time using an L-shaped dark box [15]. The extension of the detection field using a T-shaped black box equipped with two fluorescent plates enabled us to observe the two plants simultaneously (Figure 1b). The system used in this study is shown in Figure 1b. 

### 2.2. Preparation of Barley Plants with Chlorosis Symptoms and Decreased Fe Concentration in the Leaves for ^59^Fe Imaging

The molecular mechanisms underlying the acquisition of Fe are induced by Fe deficiency [17]. However, severely chlorotic leaves with necrotic spots are not suitable for evaluating the adaptive responses to Fe deficiency. In addition, the limitations of the imaging field size were considered. Barley plants were carefully prepared for ^59^Fe imaging, as described in the Materials and Methods. The soil plant analysis development (SPAD) values (chlorophyll index) of the third leaves moderately decreased in all three cultivars under Fe deficiency (Appendix A). The Fe concentration in the leaves of Fe-deficient SRB plants (Figure 2a, orange bars) was comparable to those of the other two cultivars (Figure 2b,c, orange bars), whereas fully developed Fe-sufficient leaves (L1–L3) of SRB had approximately a two-fold greater amount of Fe when compared to that of EHM (Figure 2a,b, green bars). The newest leaves of Fe-deficient SRB tended to accumulate less Fe than those of EHM and ETH (Figure 2, L4), despite the higher SPAD value (Appendix A). The Fe concentration in Fe-deficient SRB leaves for L2, L3, and L4 decreased to 33, 22, and 28% of that in the control leaves, respectively. In contrast, EHM and ETH had approximately half the concentration of Fe in the Fe-deficient leaves compared to the control. Molecular mechanisms enabling the acquisition of Fe were expected to be induced in these Fe-deficient barley plants, and these plants were deemed suitable for Fe transport imaging.

### 2.3. The Influx Rate of a Trace Amount of Fe Was Not Always Higher in Tolerant Cultivars

To examine whether the Fe-deficiency-tolerant cultivars had a great ability to localize the Fe newly acquired by roots to the newest leaves, which have a high demand for Fe to construct photosystems under Fe-deficient conditions, ^59^Fe tracer experiments were performed. Using live-autoradiography [15], we estimated the influx rate of ^59^Fe into each leaf of living barley under light conditions and thus measured the actual rate of Fe absorption (Figure 3a); 0.5 MBq of ^59^Fe as 1.24 nM Fe-citrate was applied to one plant without non-radioactive Fe. Because active Fe transport systems in roots were not induced under Fe-sufficient conditions, radioactivity in Fe-sufficient leaves was detected only at trace levels (data not shown). In the case of Fe-deficient plants, ^59^Fe influx rates were higher in the younger leaves of the three cultivars than in the older leaves (Figure 3b). However, the ^59^Fe influx rates were different among the three cultivars, in the order of EHM > ETH > SRB. The ^59^Fe influx rates of SRB were significantly lower than those of EHM (Figure 3b).

### 2.4. Downregulation Confirmation of the SUF Pathway and Accumulation of Reaction-Center Proteins

Differences in ^59^Fe influx rates among the three cultivars were observed, as were individual differences and large variances. Thus, the accumulation of reaction-center proteins on the thylakoid membrane as a major target of Fe insertion was tested to validate the influence of the decrease in Fe uptake rate.

The sulfur utilization factor (SUF) system, which generates and delivers Fe-S clusters in chloroplasts, has mainly been investigated in *Arabidopsis* [18]. The relative expression levels of *SUFB* and *GrxS14*, which are clearly downregulated by Fe deficiency, among SUF machinery-related genes [19,20] were measured. The relative expression levels of both genes were lower in the younger leaves than in the older leaves (Figure 4). The youngest leaf, L4, was premature, did not expand, and had low expression levels for both *SUFB* and *GrxS14*. *SUFB* and *GrxS14* were downregulated by Fe deficiency in the expanded leaves (L2 and L3) of all the three cultivars (Figure 4). The L3s of SRB and EHM were not significantly different because the control L3 leaf samples contained both completely expanded and emerging leaves. The relative expression levels of *SUFB* and *GrxS14* were not significantly decreased by Fe deficiency in the L4 of ETH (Figure 4e–g), and the extent of downregulation of both genes in L2 and L3 of ETH was also less than that of SRB and EHM (Figure 4).

Next, the accumulation of PsaA, PsaB, and PsaC in photosystem I (PSI), harboring the 4Fe-4S cluster supplied by the SUF system, was evaluated. The three PSI proteins decreased in the three cultivars due to Fe deficiency (Figure 5). The decreases in Psa B and PsaC were greater in SRB and ETH than in EHM. The accumulation of D1 and D2 in PSII, harboring non-heme iron, was also tested. The decreases in D1 and D2 were evident only in SRB (Figure 5). EHM maintained the amount of both PSI and PSII reaction center proteins, ETH markedly decreased PSI but not PSII, and SRB markedly decreased both PSI and PSII. Thus, the slow rate of ^59^Fe influx in SRB was consistent with the small accumulation of reaction-center proteins under Fe-deficient conditions.

## 3. Discussion

Live-autoradiography clearly demonstrated the genetic differences in the rate of Fe uptake by barley plants. The low Fe concentration (Figure 2) and small accumulation of reaction-center proteins that bind Fe in the leaves (Figure 5) of Fe-deficient SRB were consistent with its poor capability to acquire Fe from nutrient solutions containing trace amounts of Fe (Figure 3b). Our results suggest that active Fe acquisition from a low-Fe solution is not always the main factor contributing to Fe-deficiency tolerance and that the ability to acquire Fe should be directly evaluated independently of leaf color. SRB had a greater ability to recycle Fe within plants because Fe-sufficient SRB had a larger amount of Fe in leaves compared to that of the other cultivars (Figure 2). However, the newly emerging leaves of Fe-deficient SRB showed no greater Fe level than that of other cultivars. EHM may adopt the Fe-deficiency tolerant strategy due to Fe acquisition, whereas SRB may use this for sparing of Fe.

The downregulation of *SUFB* induced by Fe deficiency has also been reported in rice and cyanobacteria [21,22]. Moreover, this downregulation could be involved in the modulation of Fe allocation in *Arabidopsis* leaf cells for acclimation to low Fe levels [19]. The downregulation of genes related to the synthesis of Fe-S clusters may be caused by the depletion of Fe as a substrate for Fe-S cluster synthesis. Immediate suppression of prior Fe allocation to PSI may support the acclimation of leaves to Fe deficiency.

Plant responses to the depletion of essential elements can be categorized into three acclimation mechanisms: (1) improving nutrient uptake via roots, (2) recycling and reallocation of elements to priority organs, and (3) modulating cell physiology to survive under nutrient-limited conditions. All the steps were subjected to phenotyping. A broad survey of manganese (Mn) nutrition in barley genotypes revealed that Mn transport and maintenance of PSII are important traits for Mn deficiency tolerance [23]. Different genotypes may adopt different strategies to overcome depletion of essential elements. Phenotyping the ability to acquire essential elements will continue to be useful. Further improvement in the throughput of the imaging system enabled us to evaluate the rate of Fe uptake in approximately 20 genotypes of barley, as previously reported [11].

## 4. Materials and Methods

### 4.1. Plant Materials

Three varieties of barley: *Hordeum vulgare* L. “Sarab1” (SRB), “Ehimehadaka1” (EHM), and “Ethiopia2” (ETH) were kindly provided by Professor Kazuhiro Satoh (Barley Germplasm Center, Okayama University, Japan). The basic growth conditions were similar to those described in our previous studies [11,24]. Seedlings were grown hydroponically in a growth chamber at 24/20 °C with a light intensity of 150–200 μmol photons m^−2^ s^−1^ under a 14 h light/10 h dark cycle. Barley seeds were germinated on moist paper towels for three days and then supplemented with a half-strength nutrient solution containing 15 μM Fe-EDTA for the next three days. Plants were prepared according to the following conditions: whole shoot of each plant with an expanded third leaf can be captured within the detection field, and Fe concentration of the shoot is decreased enough to induce the Fe-acquisition pathway. Seedlings were then transferred to a standard nutrient solution (0.7 mM K_2_SO_4_, 0.1 mM KCl, 0.1 mM KH_2_PO_4_, 2 mM Ca(NO_3_)_2_, 0.5 mM MgSO_4_, 10 µM H_3_BO_3_, 0.5 µM MnSO_4_, 0.5 µM ZnSO_4_, and 0.2 µM CuSO_4_ and 10 nM (NH_4_)_6_Mo_7_O_24_), along with 30 µM Fe-EDTA for EHM and ETH, and 15 µM Fe-EDTA for SRB. When the length of the second leaf approached that of the first leaf, the plants were divided into two groups: control and Fe-deficient. Control and Fe-deficient plants were grown in standard nutrient solution and nutrient solution without Fe for four days, respectively. These plants were used for live autoradiography and gene expression analyses. The leaves were numbered in order of the developmental stage. Each leaf was segmented into two parts. One part was immediately frozen and stored at −80 °C until extraction of total RNA for quantitative PCR. The remaining part was dried and used to determine the Fe content.

An Fe-deficient culture was started during third leaf emergence to detect reaction-center proteins on the thylakoid membrane. After 16–17 days, the chlorotic leaves and corresponding control leaves were harvested, cut into fine pieces, immediately frozen in liquid nitrogen, and stored at –80 °C until extraction of total proteins.

### 4.2. Fe Determination

The dried leaves were digested in concentrated HNO_3_ at 150 °C and dissolved in 1% (*v/v*) HNO_3_. Thylakoid fractions were digested using clean reagents, instruments, and atmosphere according to the methodology described in a previous study [11] due to trace amounts of Fe in the thylakoid membranes. Fe concentration was measured using an atomic absorption spectrophotometer (AA-6300, Shimadzu, Tokyo, Japan) coupled with a graphite furnace atomizer (GFA-EX7i, Shimadzu, Tokyo, Japan).

### 4.3. Live-Autoradiography and ^59^Fe Imaging

An improved live-autoradiography system consisting of a high-sensitivity CCD camera (Hamamatsu Photonics, Hamamatsu, Japan, ImagEM X2) with a bright lens (Space Inc., Mitaka, Japan, S6 × 11), fluorescent plate, black box, and convex lens was used. The phosphor layer in the fluorescent plate, which converts β-rays into visible light, was prepared using the SiAlON phosphor powder. Two fluorescent plates were incorporated into the walls of a T-shaped black box (Figure 1b). One fluorescent plate (200 mm × 200 mm) was used to image the aboveground part of the barley plant. To reduce the noise and damage to the CCD sensor by γ-rays from ^59^Fe, lead blocks were placed at the bottom of the black box (Figure 1b).

Barley plants were fixed on the surface of the fluorescent plates as described below, and ^59^Fe tracer experiments were performed. Experiments were performed at approximately 20 °C with 40 μmol photons m^−2^ s^−1^ at the leaf surface using a household white LED lamp. Barley leaves were fixed onto the fluorescent plate using transparent vinyl strips with a thin urethane foam lining, due to the twisted orientation of the leaves (Figure 3a). This fixation method enables plants to reserve light and gas. The roots were immersed in a bag filled with 50 mL nutrient solution without Fe, covered with a lead shield, and fixed out of the fluorescent plate. We prepared ^59^Fe nutrient solution by adding 0.5 MBq of ^59^Fe-citrate (originally ^59^FeCl_3_ in 0.5 M HCl purchased from Eckert & Ziegler Isotope Products, Valencia, CA, USA and diluted with a 1% citrate solution) to 50 mL of nutrient solution without Fe. The nutrient solution without Fe was exchanged with ^59^Fe nutrient solution, and the plants were imaged immediately. The exposure time for a single image was 15 min and was continued for 4 h.

The image obtained contained several spike noises due to γ-rays, which were removed using the remove outliers function in ImageJ software (ver.1.52), a public domain software package developed by NIH, USA. Three regions of interest (ROIs) of 16 mm^2^ (4 mm × 4 mm = 5 × 5 pixels) were marked on each leaf and their average values were calculated (Figure 3a). Subsequently, time-activity curves (TACs) were generated for 59Fe Bq per 16 mm^2^ of each leaf. The ^59^Fe influx rate was calculated from the increase in radioactivity over a time window (120–210 min) when activity increased linearly.

### 4.4. Quantitative Polymerase Chain Reaction Analysis

Total RNA was isolated from leaves using an RNeasy Plant Mini Kit (Qiagen, Tokyo, Japan). cDNA was synthesized using a PrimeScript 1st-strand cDNA synthesis kit with oligo-dT primers (Takara Bio Inc., Ohtsu, Japan). Next, 1 µL of the diluted cDNA sample was subjected to quantitative real-time polymerase chain reaction (qRT-PCR) analysis in a 10 μL reaction mixture using PowerUp^TM^ SYBRTM Green Master Mix (Thermo Fisher Scientific, MA, USA). Primers specific to each gene, *SUFB and GrxS14*, which amplified cDNA but not genomic DNA (Appendix A), were used at a concentration of 0.3–0.5 μM. The PCR conditions for denaturation, annealing, and extension were 95 °C for 3 s, 60 °C for 40 s, and 72 °C for 60 s, respectively. Each sample was analyzed in triplicate. A relative quantification method, based on a standard curve, was used to calculate the results. Standard samples of the desired concentration range were prepared by combining several aliquots of cDNA samples and stepwise diluting to 1/5000. The values obtained were normalized to those of EF-1α and actin as the reference genes. The primers used in this study are listed in Appendix A.

### 4.5. Western Blot Analysis

The frozen leaves were ground into a fine powder using a mortar and pestle in liquid N_2_. The leaf powder was thoroughly suspended in protein-solubilizing buffer [25] by vortexing, and the supernatant containing total leaf proteins was recovered after centrifugation at maximum speed at 4 °C. An aliquot of the supernatant was resuspended in 80% (*v/v*) acetone and used to analyze the chlorophyll concentration [26]. Total protein with 500 ng and 50 ng chlorophyll was subjected to SDS–PAGE [25] for CBB staining and immunoblot analysis, respectively. Immunoblot analysis was performed as previously described [24]. Anti-D1, anti-D2, anti-PsaA, anti-PsaB, and anti-PsaC antibodies were obtained from Agrisera (Vännäs, Sweden).

## 5. Conclusions

To date, the ability to acquire trace elements from low-concentration solutions was assumed to be strongly correlated with tolerance to element deficiency. In the present study, 59Fe movement in a plant body under light conditions was detected using live-autoradiography. The strongly Fe-deficiency-tolerant cultivar SRB exhibited lower ability to acquire Fe than the less tolerant cultivar ETH. The results indicate the necessity of phenotype-specific testing based on experimental evidence. Therefore, new phenotyping techniques should be further developed to identify useful genetic resources.

## Figures and Tables

**Figure 1 plants-11-00817-f001:**
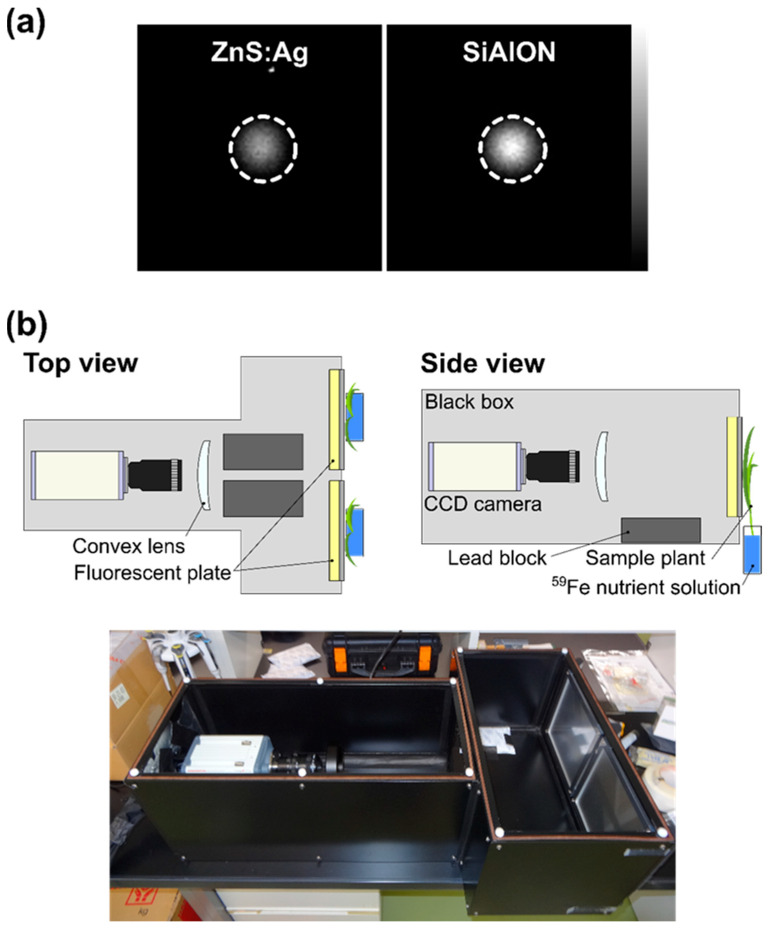
Improvement in “live-autoradiography”. (**a**) Images of a ^137^Cs point source with a diameter of 4 mm and radioactivity of 100 kBq acquired over 5 min using ZnS:Ag (left) and SiAlON (right) scintillators. The dashed circles in the figure indicate ROIs. (**b**) Schematic of improved live-autoradiography system (top view on the left, side view on the right) and photo of live-autoradiography system.

**Figure 2 plants-11-00817-f002:**
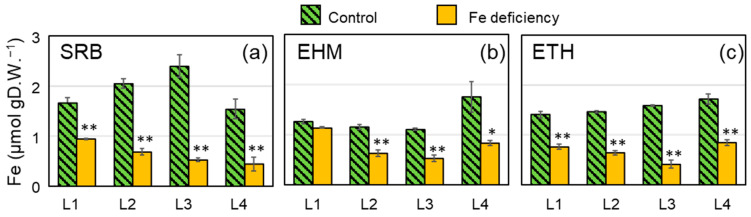
Total Fe concentration in each leaf of seedlings after four days of Fe depletion. Fe contents were evaluated in three barley cultivars grown in hydroponic solutions containing 30 μM (green) or 0 μM (orange) EDTA-Fe for four days. (**a**) “Sarab1” (SRB), (**b**) “Ehimehadaka1” (EHM), and (**c**) “Ethiopia 2” (ETH). Values represent the mean (SE) of the three plants. L1 is the oldest leaf and L4 is the premature leaf just emerging. * *p* < 0.05 and ** *p* < 0.01 indicate significant differences (according to Student’s *t*-test) between the control leaf and Fe-deficient leaf.

**Figure 3 plants-11-00817-f003:**
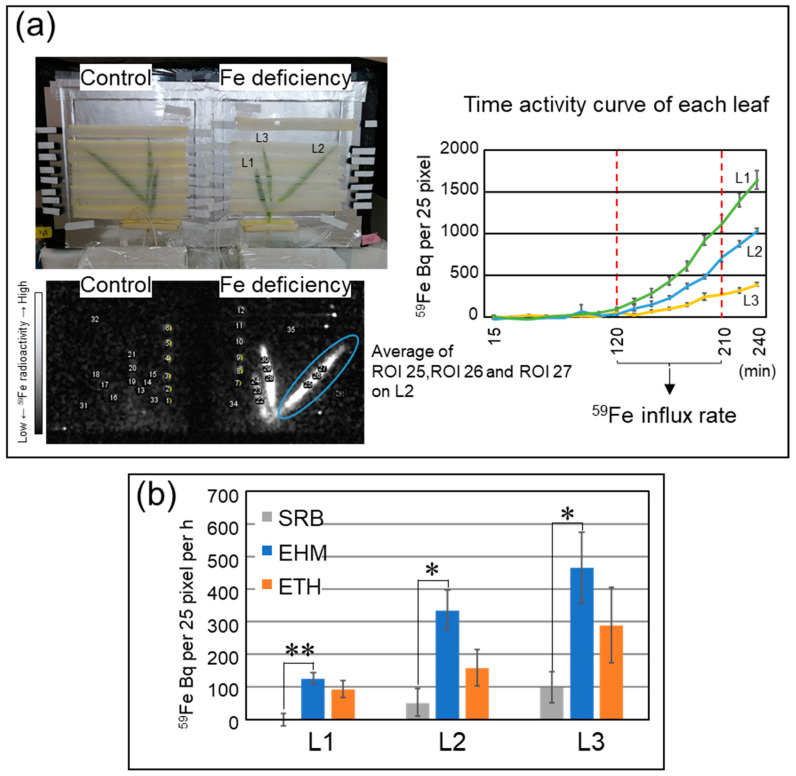
^59^Fe influx rate of Fe-deficient leaves detected by live-autoradiography. (**a**) Scheme for calculation of ^59^Fe influx rate to leaves. Three regions of interest (ROIs) of 16 mm^2^ (4 mm × 4 mm = 5 × 5 pixels) were set for each leaf, and average values of the three ROIs were calculated. Time–activity curves (TACs) for ^59^Fe Bq per 16 mm^2^ of each leaf were then generated. ^59^Fe influx rate was calculated from the increase of radioactivity during a time window (120–210 min) when activity was linearly increased. (**b**) ^59^Fe influx rate of each leaf was measured using three plants of each cultivar. Each TAC is represented in Appendix A. Only Fe-deficient leaves are represented, since the ^59^Fe influx rates of control leaves were slow and sometimes undetectable. Gray bars, SRB; blue bars, EHM; orange bars, ETH. * *p* < 0.05 and ** *p* < 0.01 indicate significant differences (according to Student’s *t*-test) between the two cultivars.

**Figure 4 plants-11-00817-f004:**
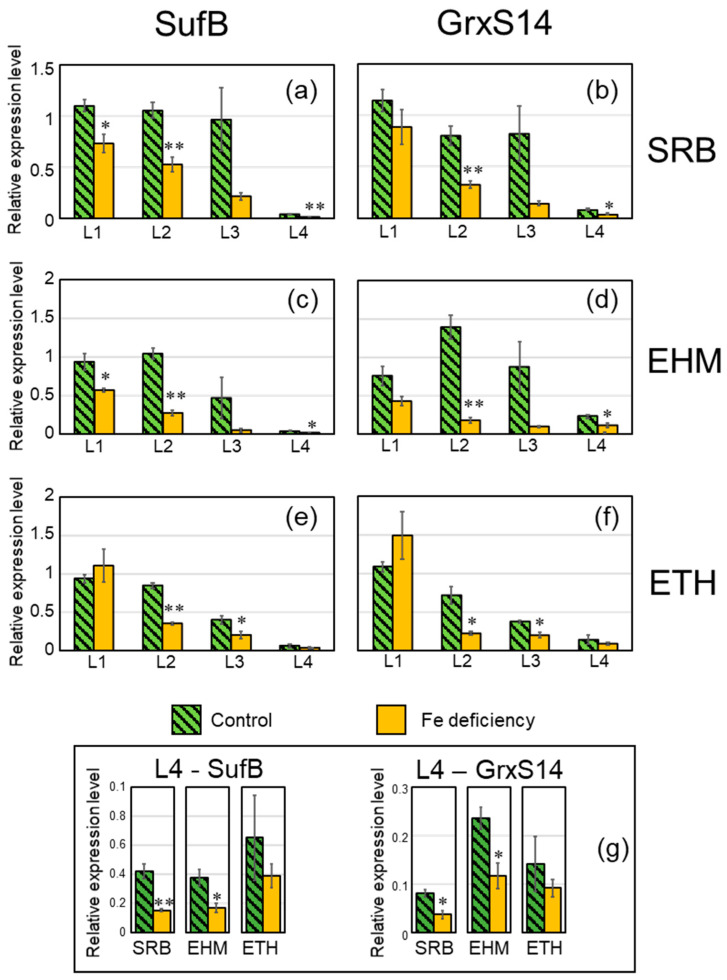
Downregulation of *SUFB* and *GrxS14* at day four after the onset of Fe depletion. Relative expression values were obtained from qRT-PCR performed on total cDNA of leaf samples collected from the same plants as those of Figure 1. Each sample of the three biological replicates was subjected to triplicate experiments, and expression levels of *SUFB* and *GrxS14* were normalized against those of *EF-1α* and *actin*. Error bars indicate the standard error of the three biological replicates. * *p* < 0.05 and ** *p* < 0.01 indicate significant differences (according to Student’s *t*-test) between the control leaf and Fe-deficient leaf. (**a**,**b**) SRB, (**c**,**d**) EHM, and (**e**,**f**) ETH. (**a**,**c**,**e**) *SUFB* and (**b**,**d**,**f**) *GrxS14*. Enlarged L4 data are shown in panel (**g**).

**Figure 5 plants-11-00817-f005:**
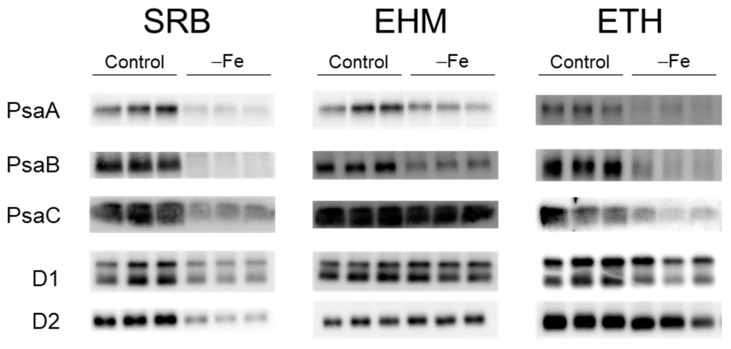
The decrease in the accumulation of reaction-center proteins under the Fe-deficient condition. PsaA, PsaB, and PsaC of PSI reaction-center and D1 and D2 of PSII reaction-center were detected using each specific antibody. Each lane was loaded with 50 μg of chlorophyll. A set of three lanes means three independent plants. Images of CBB staining are shown in Appendix A.

## Data Availability

Not applicable.

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
