# Peer review of "“Live-Autoradiography” Technique Reveals Genetic Variation in the Rate of Fe Uptake by Barley Cultivars"

_plants, 2022, doi:10.3390/plants11060817_

Round 1
Reviewer 1 Report
The manuscript deals with the application of a live-autoradiography technique in revealing genetic variation in the rate of Fe uptake by barley cultivars – as the title precisely describes the goal and, in the same time, the message of this work. This work is a proper contribution to the special issue „From Phenotyping to Phenomics—Techniques for Exploring Plant Traits and Diversity”.
The text is easy to read, the methodologies are appropriate and the presentation of the results are good.
I have only minor concerns prior to publication:
Results 2.1, line 89: Here, I think, a very short intro and a reference to the original system is needed as the paragraph starts very much "in medias res".
line 128: were / had been both remained in the text – please correct
Figure 3. Indication of color codes is missing from Fig. 3a. – although this information is not directly necessary for understanding and it can be figured out but it may be added as it would help faster orientation in the figure.
line 161: I suggest writing orange bars instead of red bars as it more precisely describes the color, I think.
Author Response
We sincerely appreciate the careful comments of the reviewers to improve the manuscript.
Please refer to the attached revised manuscript.
We revised the manuscript after taking into consideration the comments of all reviewers.
Revisions are indicated in red font.

Reviewer 2 Report
The manuscript deals with the "live-autoradiography technique for monitoring the rate of Fe uptake by different genotypes of barley". In overall, manuscript has been written well; however, it needs some revisions before considering for publication.
1. Write keywords alphabetically.
2. Avoid direct quote of others. For instance:
"Jahnke et al. [2] successfully visualized carbon dy-36 namics in plants using the PlanTIS system."
"Streum et al. [3] further developed ”phenoPET” 37 for high-throughput phenotyping."
"Ruwanpathirana et al. [5] exploited clinical PET to monitor 22Na transport in plants, and Brezovcsik et al. [6] compared 52Mn uptake and transport in two maize genotypes using a mini-PET camera."
3. Page 8, Line 239; "with light intensity of 150–200 μmol photons m−2 s−1 under a 14 h light/10 h dark cycle" Support it by references.
4. Page 8, Line 242; "We prepared..." Avoid to use several pronouns in a scientific paper!!
Line 16 "We adopted...."; Line 20 "Our observation..."; Line 38 "We visualized..." Line 50 "We recently reported..."; Line 74 "We compared the rate..." Line 76 "We used two..." Line 77 We present that..." Line 89 "We used SiAlON..." Line 114 "we considered the limitations...". Line 115 "We carefully prepared..."; Line 139 "We investigated whether..."; Line 143 "We applied 0.5 MBq..."; Line 165 "We observed differences..."; Line 165 "Thus, we tested the accumulation..."; Line 170 "We measured the relative...." and so on.....
5. Write a conclusion
Author Response

(The authors gave the same response as above.)

Round 2
Reviewer 2 Report
Reviewers’ comments have been addressed
Author Response
Please refer attached manuscript.
